# The Influence of Hydrogen Passivation on Conductive Properties of Graphene Nanomesh—Prospect Material for Carbon Nanotubes Growing

**Vladislav V. Shunaev [1] and Olga E. Glukhova [1,2,*]**

[1] Department of Physics, Saratov State University, 410012 Saratov, Russia; shunaevvv@sgu.ru
[2] Institute for Bionic Technologies and Engineering, I. M. Sechenov First Moscow State Medical University (Sechenov University), 119991 Moscow, Russia
* Correspondence: glukhovaoe@sgu.ru; Tel.: +7-8452-514562

**Abstract:** Graphene nanomesh (GNM) is one of the most intensively studied materials today. Chemical activity of atoms near GNM's nanoholes provides favorable adsorption of different atoms and molecules, besides that, GNM is a prospect material for growing carbon nanotubes (CNTs) on its surface. This study calculates the dependence of CNT's growing parameters on the geometrical form of a nanohole. It was determined by the original methodic that the CNT's growing from circle nanoholes was the most energetically favorable. Another attractive property of GNM is a tunable gap in its band structure that depends on GNM's topology. It is found by quantum chemical methods that the passivation of dangling bonds near the hole of hydrogen atoms decreases the conductance of the structure by 2–3.5 times. Controlling the GNM's conductance may be an important tool for its application in nanoelectronics.

**Keywords:** graphene nanomesh; carbon nanotube; mathematical modeling; growing; transmission

## 1. Introduction

GNM, also called a "holey graphene", is one of the most intensively studied carbon nanostructure. It combines outstanding properties of graphene and unique physical and chemical properties determined by the edge atoms around nanoholes [1,2]. In recent years, many strategies of GNM synthesis have been developed: solvothermal reaction [3], non-templated synthesis [4], self-templated synthesis [5], single-step air oxidation process [6], etc. [7]. GNM demonstrates a band gap (BG) that can be tuned by changing the neck's width (the distance between neighbor holes) and nanohole's topology [8–11]. It was shown that a field-effect transistor on the base of GNM surpasses its graphene analogue in terms of output conductance, voltage gain, and maximum oscillation frequency [12]. Since dangling atoms of GNM are chemically active, it is energetically favorable to decorate GNM with different types of atoms and molecules—for example, H, F, N, DNA, $NO_2$, and $NH_3$ [13–16]. The sensibility of GNM properties to adsorbed molecules makes this material a prospect element for electrochemical sensing of environmental toxins [17]. The density functional theory (DFT) calculations established that a BG of GNM passivated by hydrogen varied in the range of 0.35–0.95 eV in dependence on a nanohole's size and hydrogen concentration [13]. Though the influence of the neck's width on the conductive properties of GNM were estimated [4], the influence of hydrogen passivation isn't defined yet. To our mind, this research could significantly enlarge GNM knowledge and expand an area of GNM application.

Another important advantage of GNM is the possibility of CNT growing on its surface. The obtained hybrid composite known as pillared graphene structures supported by vertically aligned CNTs (VACNT-graphene) has found its application in hydrogen storage, nanoelectronic devices, supercapacitors, and biosensors [18–20]. We have already simulated

the growth of armchair and chiral CNTs on GNM surface with a hole in the form of a hexagon [21,22] and found the neck's width that provided the parameters of the most favorable growing. It's obvious that the geometric form of the hole will influence this process and it will change the conductive parameters of GNM. Thus, the goal of this paper is to determine the influence of the nanohole's form and hydrogen's passivation of this hole on conductive properties of GNM. Additionally, the comparison of CNT growing in nanoholes of different forms will be performed in terms of energy efficiency.

## 2. Methods and Results

### 2.1. Atomic Structures of GNM with Holes of Different Geometry

We considered GNM's atomic structure with holes of four geometric forms: hexagonal, circle, square, and triangle. These holes were taken from [23] and were selected in such a way that their areas were approximately the same (Figure 1). The atomic cells of studied GNM were optimized by the self-consistent-charge density-functional-tight-binding (SCC DFTB) method [24] in the basic pbc-0-3 set developed for organic molecules [25] in a periodic box with translation vectors $L_x$ = 35.161 Å, $L_y$ = 34.371 Å where X axis corresponded to the zigzag direction and Y axisto the armchair one. All calculations were performed in DFTB + 21.2 [26] free software at 300 K package with the max force component = $1e^{-4}$. We showed earlier that the CNT growing from GNM with these translation vectors was the most favorable [22]. The lowest energy matches to the structure with the hexagonal hole (−47.23 eV/atom). GNM with holes of circle and square forms have almost similar energy (−47.19 and −47.17 eV/atom), while GNM with the triangle hole has the largest energy (−47.06 eV/atom). The more energy per atom, the less stable the structure is. Thus, the presence of GNM with triangle holes is unlikely in practice: such a hole will probably be filled with carbon atoms or with atoms of other chemical elements. So, it is probable that GNM with the triangle hole will be completed to GNM with another geometrical form of the VACNT-graphene composites.

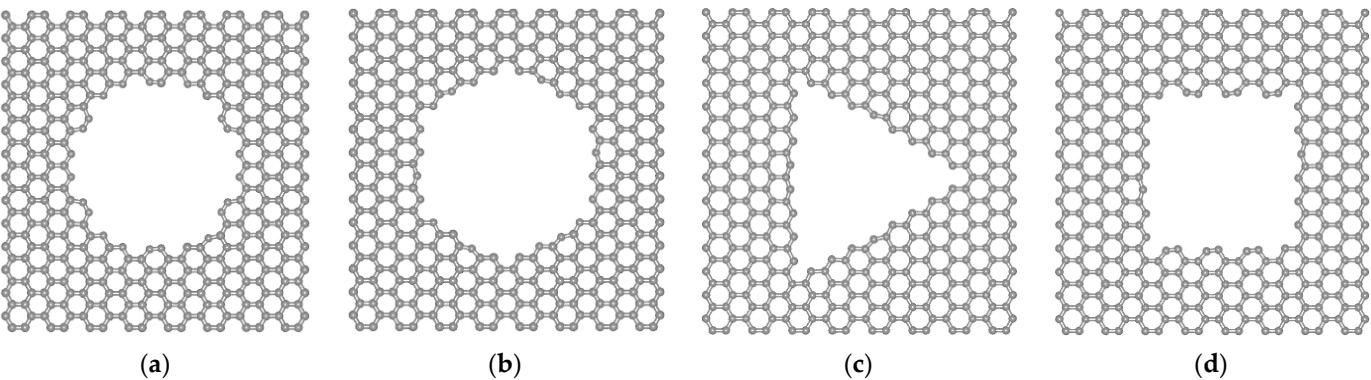

(**a**)　　　　　　(**b**)　　　　　　(**c**)　　　　　　(**d**)

**Figure 1.** GNM's atomic structures with holes of different geometric forms: (**a**) circle; (**b**) hexagon; (**c**) triangle; (**d**) square. The areas of holes are almost similar and equal to the area of 55 hexagons (5.14 Å$^2$). All structures were optimized by the SCC-DFTB method in a periodic box with translation vectors $L_x$ = 35.161 Å along the zigzag direction and $L_y$ = 34.371 Å along the armchair direction.

### 2.2. Conductive Properties of GNM

Since GNM with circle and hexagonal holes are the most energetically favorable structures next calculations were performed for these atomic cells. To analyze conductive properties of the considered structures, transmission functions were calculated with application of the non-equilibrium Green-Keldysh function method [27] and the Landauer–Büttiker formalism that allowed the studying of quantum transport of electrons taking into account the elastic scattering of electrons in inhomogeneities [28]:

$$T(E) = Tr\left(\Gamma_S(E)G_C^A(E)\Gamma_D(E)G_C^R(E)\right) \tag{1}$$

where $G_C^A(E)$, $G_C^R(E)$ are the advanced and retarded Green matrices that describe the contact of an object with an electrodes, $\Gamma_S(E)$, $\Gamma_D(E)$ are the level broadening matrices for the source and drain, respectively. Transmission functions of GNM with circle and hexagonal holes, pure and passivated by hydrogen, are shown in Figure 2. Dimension of transmission function $T(E)$ is quantum of conductance—$e^2/2\,h$ (value of the only transmission channel). BG $E_g$ for GNM with the circle hole is 0.14 eV, for GNM with hexagonal form—0.12 eV (Figure 2a,b). The addition of hydrogen expectedly increased the BG: to 0.21 eV for GNM with the circle hole and to 0.17 eV for GNM with the hexagonal hole (Figure 2c,d). Local peaks on transmission functions correspond to $T(E) = 0.5$ when current flows along the armchair direction and to $T(E) = 1$ when current flows along the zigzag one. More stepped character of transmission functions in the case of passivated edges can be explained by features of the SCC-DFTB method sensible to structures with atoms with dangling bonds.

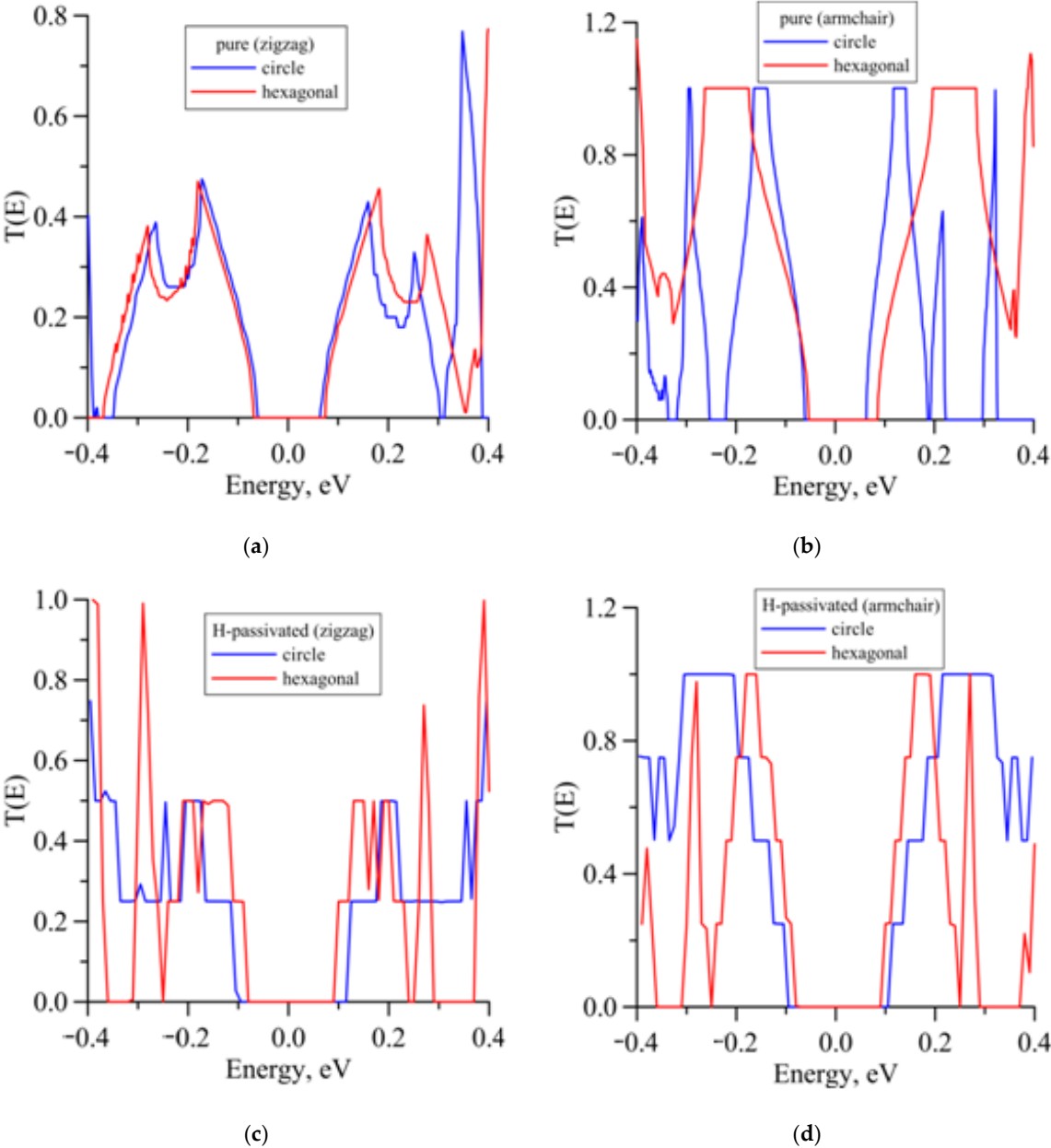

**Figure 2.** Transmission functions of GNM: (**a**) with circle and hexagonal holes along the zigzag direction; (**b**) with circle and hexagonal holes along the armchair direction; (**c**) with circle and hexagonal holes passivated by hydrogen along the zigzag direction; (**d**) with circle and hexagonal holes passivated by hydrogen along the armchair direction.

The electric conductivity $G$ was calculated on the base of transmission function by the formula:

$$G = 2e^2/h \int_{-\infty}^{+\infty} T(E)F_T(E - E_F)dE \tag{2}$$

where $E_F$—Fermi energy of contacts material; $e$—electron's charge; $h$—Planck's constant; $F_T$—function that determines the value of the temperature broadening. The values of conductivity and resistance are shown in Table 1. The table demonstrates that the biggest conductivity corresponds to GNM with the hexagonal hole (5.16 µS in the armchair direction and 3.21 µS in the zigzag direction). The biggest resistance corresponds to GNM with circle holes passivated by hydrogen (1.09 MOhm in the armchair direction and 2.04 MOhm in the zigzag direction).

**Table 1.** The values of the BG, conductivity and resistance for GNM with circle and hexagonal holes.

| Current Direction | Armchair | | | | Zigzag | | | |
|---|---|---|---|---|---|---|---|---|
| Hole's Form | Circle | Circle | Hexagon | Hexagon | Circle | Circle | Hexagon | Hexagon |
| Passivation | Pure | H | Pure | H | Pure | H | Pure | H |
| $E_g$, eV | 0.14 | 0.21 | 0.12 | 0.17 | 0.14 | 0.21 | 0.12 | 0.17 |
| G, µS | 3.16 | 0.91 | 5.16 | 1.99 | 1.92 | 0.49 | 3.21 | 1.46 |
| R, MOhm | 0.32 | 1.09 | 0.19 | 0.50 | 0.52 | 2.04 | 0.31 | 0.68 |

*2.3. Virtual Growing of VACNT(11,10)-Graphene*

We chose CNT (11,10) for growing since it had the smallest radius from the most often synthesized CNTs [29–31]. Merging CNT and GNM was performed by the original methodic that provides building seamless junctions of carbon nanostructures [32]. The process of jointing was carried out in three stages (Figure 3). At first, we selected edge atoms with two covalent bonds in atomic structures of CNT and GNM (red and blue atoms in Figure 3a). Then, the special algorithm randomly generated new atoms between selection (Figure 3b). At the final stage the obtained atomic structure was refined by the SCC-DFTB method. Note that according to this methodic, a various number of final supercells may be obtained. It was demonstrated earlier that the atomic mesh VACNT-graphene with the lowest energy corresponded to the case with the least number of pentagons and hexagons in a seam field [22,33]. The most energetically favorable atomic structure VACNT(11,10)-graphene had only heptagons in its seam field (Figure 3c). For comparison, one of the atomic structures obtained during numerical experiments with several pentagons and even an octagon is shown in Figure 3d—its energy exceeds the energy of the optimal structure by 0.18 eV/atom.

Next, we calculated the energy profiles of the growing of the obtained VACNT(11,10)-graphene supercells according to the «virtual growing» methodic [21,22]. The essence of this technique is to calculate the energy after the addition of each new layer. Energy profiles of the growing of the VACNT(11,10)-graphene supercells with initial cell of GNM with holes of different geometrical forms are shown in Figure 4. The «0» at the abscess axis corresponds to the initial cell. As it seen from Figure 4, the lowest energy matches to the structure with the hexagonal hole (−47.23 eV/atom). GNM with holes of circle and square forms have almost similar energy (−47.19 and −47.17 eV/atoms) while GNM with the triangle hole has the biggest energy (−47.07 eV/atom). Thus, the presence of GNM with triangle holes is unlikely in practice: such a hole will probably be completed to holes of other geometric shapes or filled with atoms of other chemical elements. The energy profiles of VACNT(11,10)-graphene have a similar view. At the initial stage, the energetic minimum is observed (for the hexagonal hole when N = 2, for the circle, square, and triangle when N = 4), then—the local maximum (for hexagonal hole when N = 4, for the circle and square when N = 6, for the triangle when N = 8), and the curves decrease when the number of layers increase. The difference between energies at the local maximum

and minimum corresponds to the energy barrier that should be overcome for further CNT's growing. The lowest energy barrier (0.014 eV/atom) corresponds to GNM with the circle hole, so the growing of VACNT(11,10)-graphene from GNM with the circle hole is the least energy-consuming. For comparison, the energy barrier in the case of the circle and square forms equals 0.021 eV/atom, in the case of the triangle hole—0.016 eV/atom. After overcoming the energy barrier, the energy per atom falls with growth in the number of layers and the form of the initial hole stops influencing the composite's growth. Note that the energy per atom values after the growing of 10 layers are almost similar in all cases.

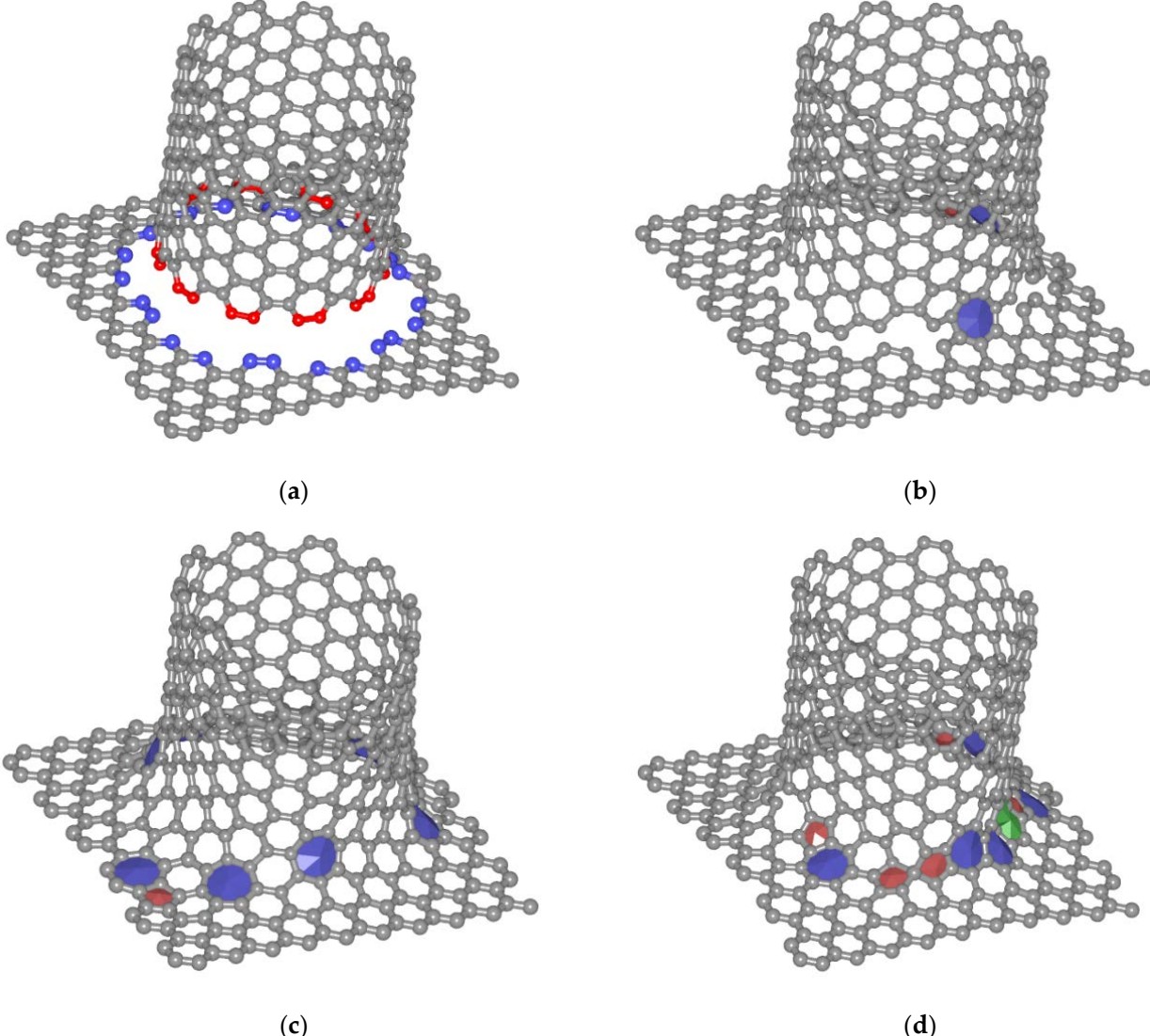

(**a**)                                                                        (**b**)

(**c**)                                                                        (**d**)

**Figure 3.** The growing of VACNT(11,10)-graphene from GNM with the circle hole: (**a**) selection of the edge atoms in a seam field; (**b**) generation of random atoms in the space between GNM and CNT; (**c**) the most energetically favorable atomic structure VACNT(11,10)-graphene; (**d**) one of the VACNT(11,10)-graphene's atomic structure variants with energy that exceeds the variant in Figure 3c by 0.18 eV/atom. The red color corresponds to pentagons, blue—to heptagons, green—to octagon.

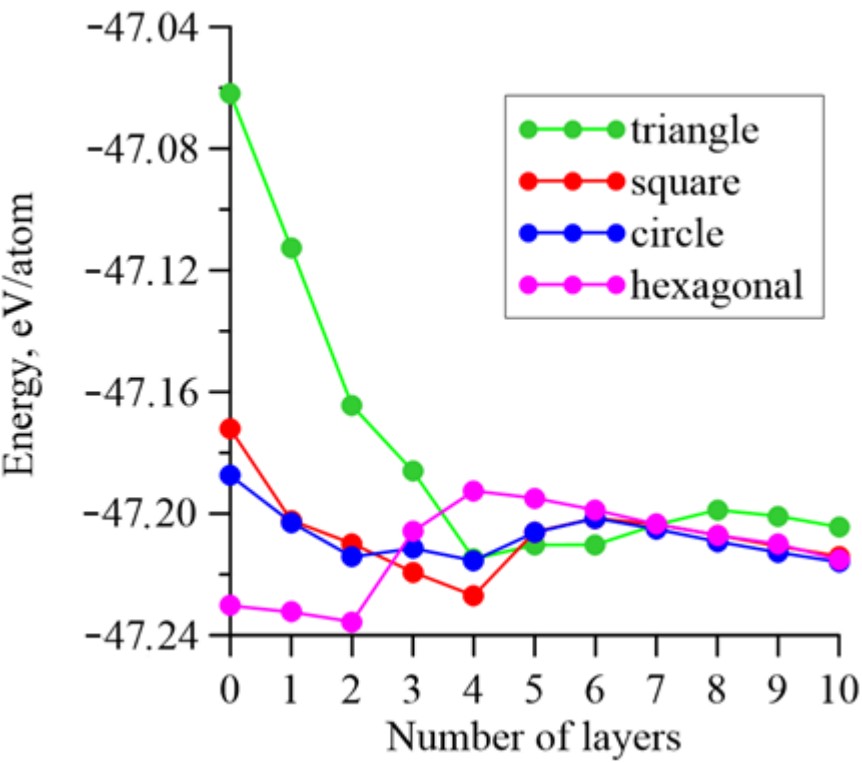

**Figure 4.** Energy profiles of the growing of the VACNT(11,10)-graphene supercells with the initial cell of GNM with holes of different geometrical forms.

## 3. Conclusions

By the application of the SCC-DFTB method, the supercells of GNM with nanoholes of four different geometric forms were established. The minimum energy corresponded to GNM with the circle form, the maximum—to GNM with the triangle form. Thus, the presence of GNM with the triangle hole is highly improbable, it will be completed to holes of other geometric forms or will be filled by atoms of other types. By the original methodic, the virtual growing of CNT (11,10) from the obtained GNM supercells was performed. It was found that the growing of VACNT(11,10)-graphene from GNM with the circle form is the most energetically favorable since it has the lowest energy barrier 0.014 eV/atom. The calculation of transmission function revealed that the passivation of holes' atoms by hydrogen enlarged the BG from 0.12 to 0.17 eV in the case of the hexagonal hole, from 0.14 to 0.21 eV—in the case of the circle hole. The obtained results demonstrate that hydrogenation of a hole can be an effective tool for tuning the conductive properties of GNM that can be used in various nanoelectronics applications such as field-effect transistors.

**Author Contributions:** Conceptualization, O.E.G. and V.V.S.; methodology, O.E.G.; software, V.V.S.; validation, O.E.G. and V.V.S.; formal analysis, O.E.G. and V.V.S.; investigation, O.E.G. and V.V.S.; resources, O.E.G.; data curation, O.E.G.; writing—original draft preparation, V.V.S.; writing—review and editing, O.E.G. and V.V.S.; visualization, V.V.S.; supervision, O.E.G.; project administration, O.E.G.; funding acquisition, O.E.G. All authors have read and agreed to the published version of the manuscript.

**Funding:** This research was funded by the Ministry of Science and Higher Education of the Russian Federation (project no. FSRR-2020-0004).

**Institutional Review Board Statement:** Not applicable.

**Informed Consent Statement:** Not applicable.

**Data Availability Statement:** Not applicable.

**Conflicts of Interest:** The authors declare no conflict of interest.

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
