# Peer review of "The Influence of Hydrogen Passivation on Conductive Properties of Graphene Nanomesh—Prospect Material for Carbon Nanotubes Growing"

_carbon, 2021_

Round 1

Reviewer 1 Report

In this work, the authors mention that they want to study graphene nanomesh (GNM), where they aim to determine the influence of the graphene nanohole’s form and hydrogen’s passivation of the holes' edges on the conductive properties of GNM. The authors also the study the effect of the CNT growing in nanoholes of different forms to determine from the view of energy benefit will be performed.

  • Although the idea of studying the shape of the holes in graphene layer and their passivation is not new, the authors might have interesting results, however the presentation of the results need to be clearer. 
  • The authors could have started by showing the results of simply passivating the holes with hydrogen atoms, before  the growing of CNT. It wasn't clear to me who they chose to start with the effect of the CNT growth. It would be important to compare the results of the simple passivation with the obtained results. 
  • Also the authors have to clarify more on this sentence (lines 98-99): "....Thus, the presence of GNM with triangle holes is unlikely in practice: probably, such a hole will be completed to holes of other 99 geometric shapes or filled with atoms of other chemical elements".
  • In Figure 3, the energy/atom vs number of layers is almost the same after the fourth layer, including the triangle shape, the authors need to comment on that.
  • Control calculations are needed, e.g. passivation with other atoms, such as oxygen or nitrogen. 
  • The language needs to be revised.

Author Response

We thank Referee for his kind notes and questions. We believe that the paper was improved.

  • The authors could have started by showing the results of simply passivating the holes with hydrogen atoms, before  the growing of CNT. It wasn't clear to me who they chose to start with the effect of the CNT growth. It would be important to compare the results of the simple passivation with the obtained results. 

The structure of the Section 2 was reorganized according to desire of Referee. In 2.1 we showed atomic structures of GNM with holes of different forms, in 2.2 – the influence of passivation on conductive properties of GNM with circle and hexagonal hole, in 2.3 – the growing of VACNT from GNM with different holes of different forms

  • Also the authors have to clarify more on this sentence (lines 98-99): "....Thus, the presence of GNM with triangle holes is unlikely in practice: probably, such a hole will be completed to holes of other 99 geometric shapes or filled with atoms of other chemical elements".

The less energy of atomic structure the more stable it is. The highest energy per atom corresponds to GNM with triangle hole (-47.06 eV/atom) that exceeds the energy of the most favorable GNM with hexagonal hole on 0.16 eV! This indicates that GNM with triangle hole is not stable and its atomic structure will easily absorb carbon atoms as well as atoms of another type. So, it’s probable the GNM with triangle hole will be built to GNM with holes of another types or it will be favorable to grow VACNT on its surface.

The explanation was added to the text (Section 2.1)

  • In Figure 3, the energy/atom vs number of layers is almost the same after the fourth layer, including the triangle shape, the authors need to comment on that.

The key moment on VACNT’s growing is overcoming of energy barrier that is mentioned in the text of the paper (0.014 eV/atom corresponds to GNM with circle hole, 0.021 eV/atom – to the case of circle and square hole,  in the case of triangle hole – 0.016 eV/atom). After overcoming of this barrier the energy per atom falls with increasing of layer numbers that indicates the increase of energy stability of considered structures. So the form of initial hole stops to influence in composite’s growth.

The explanation was added to the text (Section 2.3)

  • Control calculations are needed, e.g. passivation with other atoms, such as oxygen or nitrogen. 

Unfortunately, the optimization as well as calculation of transmission function for many-atoms structure (and GNM is one of them) by quantum methods is time-consuming process. For the given 10 days we are unable to simulate passivation by oxygen or nitrogen. But our results match DFT calculations [Ref.13] : the addition of hydrogen leads to increase of energy gap.

  • The language needs to be revised.

The language was revised.

Reviewer 2 Report

In this manuscript, the authors study two problems that relate to graphene nanomeshes (GNMs). First, they use the self-consistent charge density-functional tight-binding (SCC DFTB) method to study the growth of carbon nanotubes (CNTs) from within the active pores of GNMs.  They demonstrate the procedure by utilizing 4 pore geometries to grow a (11,10) CNT. They report the energies of various growth stages.

The authors also study the conductance of GNMs (unpassivated and hydrogen-passivated). The compute the transmission functions in a Landauer Buttiker framework using the Keldysh formalism that utilizes non-equilibrium Green's functions. They use that to calculate the conductance of a few GNM systems in the armchair and zigzag directions.

I recommend the acceptance of the manuscript after the following points are addressed (I do not request to see the revised manuscript)

1- Figure 4 shows the transmission functions for unpassivated GNMs with circular and hexagonal pores. Previous work showed that unpassivated GNMs suffer from midgap states localized at the pore-edge atoms (please see supplementary information of 

https://pubs.acs.org/doi/pdf/10.1021/ct4000636

The authors should explain the discrepancy between their findings and previously published results.

2- In Fig.3, it is unnecessary to cut the y-axis and shift it at "-47.16".

3- The authors should mention the name of the DFTB package used, and any relevant computational parameters (cutoffs, ...etc)

4- The language of the manuscript must to be revised. The text is mostly comprehensible to a native speaker, but a thorough revision will make the flow much smoother.

Author Response

We thank Referee for his kind notes! We tried to answer the questions and improved paper according to his suggestions.

  1. Figure 4 shows the transmission functions for unpassivated GNMs with circular and hexagonal pores. Previous work showed that unpassivated GNMs suffer from midgap states localized at the pore-edge atoms (please see supplementary information of 

https://pubs.acs.org/doi/pdf/10.1021/ct4000636

The authors should explain the discrepancy between their findings and previously published results.

In the indicated article, it is possible to observe an energy gap on the DOS graph for both non-passivated and hydrogen-passivated GNMs. However, there are no contradictions with our article. As indicated in Table 1 of our article, the unpassivated GNM had an energy gap of 0.12-0.14 eV depending on the geometric shape of the hole, while the addition of hydrogen atoms lead to an increase in the gap to 0.17-0.21 eV.

  1. In Fig.3, it is unnecessary to cut the y-axis and shift it at "-47.16".

The figure was changed (in new version it’s Fig.4)

  1. The authors should mention the name of the DFTB package used, and any relevant computational parameters (cutoffs, ...etc)

The following text was addes: «All calculations were performed in DFTB+ 21.2 [26] free software at 300 K package with max force component = 1e-4

4- The language of the manuscript must to be revised. The text is mostly comprehensible to a native speaker, but a thorough revision will make the flow much smoother.

The language was revised.

Round 2

Reviewer 1 Report

The authors have addressed the comments and the added parts in the revised manuscript made it clearer. I have no further comments.